# Reduction of Ischemic Stroke Associated Disability in the Population: A State-Wide Stroke Registry Analysis over a Decade

**DOI:** 10.3390/jcm11236942

**Published:** 2022-11-25

**Authors:** Christian Foerch, Martin A. Schaller-Paule, Helmuth Steinmetz, Björn Misselwitz, Ferdinand O. Bohmann

**Affiliations:** 1Department of Neurology, University Hospital Frankfurt, Goethe-University, 60528 Frankfurt am Main, Germany; 2Department of Psychiatry and Psychotherapy, University Medical Center of the Johannes Gutenberg—University Mainz, 55131 Mainz, Germany; 3Institute of Quality Assurance, 65760 Eschborn, Germany

**Keywords:** prevention, incidence, ischemic stroke, health care, registries, anticoagulants

## Abstract

(Background): Effective prevention strategies and acute therapies have been established and distributed in recent years to reduce the global burden of stroke. However, beyond randomized clinical trials, limited data exist on the real-world impact of these measures. Our goal was to analyze whether the stroke-associated disability in the population decreased over time based on a state-wide stroke registry analysis. (Methods): Consecutive data from a state-wide inpatient stroke registry covering the entire federal state of Hesse, Germany, were obtained. The clinical data of 141,287 patients with ischemic stroke (ICD-10: I63) admitted between 2010 and 2019 were included. The primary outcome was the odds ratio for a change of modified Ranking Scale (mRS) at discharge over time, estimated by ordinary logistic regression and adjusted for age and sex. The secondary outcome was the odds ratio for a lower National Institutes of Health Stroke Scale (NIHSS) score at hospital admission. (Results): The absolute number of severely disabled (mRS 4–5) stroke patients at discharge decreased over time (2010: 3223 (equivalent to 53/100,000 population); 2019: 2429 [39/100,000 population]). The odds at hospital admission for a higher mRS at discharge decreased significantly by 3.7% per year (OR 0.963 (95% CI 0.960–0.966), *p* < 0.001). The absolute number of severely affected stroke patients (NIHSS > 15) at admission declined over time (2010: 1589 [26/100,000]; 2019: 1185 [19/100,000]; *p* < 0.001). The odds for a higher NIHSS score at admission to hospital decreased by 3.8% per year (OR 0.962 (95% CI 0.959–0.965), *p* < 0.001). Trends were most prominent for patients aged 80 years and older and for patients with atrial fibrillation but absent in patients <60 years. (Conclusions): Stroke-associated disability in the population steadily decreased between 2010 and 2019. The improved prevention of severe strokes in elderly patients may be a major driver of this observation.

## 1. Introduction

The primary target of all efforts in stroke prevention and acute care is to reduce stroke-associated disability in the population. Reducing the burden of stroke on healthcare systems with limited resources is vital, as both the incidence of stroke, as well as prevalence of post-stroke patients, are going to increase due to an aging western population [1]. To date, oral anticoagulants are widely applied to patients with atrial fibrillation for the prevention of cardioembolic strokes, and statins and antiplatelets are broadly used to reduce the risk of atherothrombotic cerebrovascular events [2]. Non–vitamin K antagonist oral anticoagulants have proven to be safe and effective options for the treatment of nonvalvular atrial fibrillation and have been increasingly prescribed since 2009 [3,4,5]. In the case of an ischemic stroke, patients increasingly more often undergo recanalizing treatment, including intravenous thrombolysis and endovascular thrombectomy. In randomized clinical trials, all of these acute measures proved highly effective [6,7]. Over the past decade, successive implementation of these improvements in acute care and stroke prevention into routine clinical practice may have contributed to reducing the burden of stroke in the population. Prior retrospective analysis of healthcare data from China, Japan, and Israel, ranging from the year 2002 to 2016, 2010 to 2016, and 2004 to 2013, respectively, described a decline in disability after stroke but were conducted prior to the widely available establishment of mechanical recanalization [8,9,10]. This study aims to determine whether the burden of stroke on the healthcare system in the state of Hesse, Germany, has declined over the past decade using population-based health data, with a focus on the evolution of stroke severity at hospital admission and stroke disability at patient discharge. The goal of this study was, therefore, to assess whether the stroke-associated disability in the population decreased over time based on a state-wide stroke registry analysis.

## 2. Materials and Methods

We obtained data from a large quality assurance inpatient stroke registry covering the entire federal state of Hesse in Germany (GQH, Geschäftsstelle Qualitätssicherung Hessen). Data entry is compulsory as a quality assurance measure by federal law and clinical data is collected and entered by the responsible stroke physicians. Since 2005, more than 95% of all hospitalized strokes in the state have been covered and periodically verified by administrative hospital data (for detailed information visit: http://www.gqhnet.de, accessed on 12 September 2022). Informed written patient consent was not required before patient enrolment into this registry. We included all patients with ischemic stroke (ICD-10: I63) admitted in ten subsequent years (2010–2019). Recorded variables include age, sex, vascular risk factors, oral anticoagulation therapy at admission, National Institutes of Health Stroke Scale (NIHSS) at admission, thrombolysis, thrombectomy, length of hospital stay, stroke etiology (according to TOAST), and functional outcome according to the modified Rankin Scale (mRS) at hospital discharge. Patients transferred between inner-state hospitals within 24 h (*n* = 2547) were excluded. Though patient data entry was performed prospectively, this is a retrospective study because it was designed post data acquisition.

### Statistical Analysis

The primary outcome variable was the common odds ratio for a change of mRS at discharge over time, estimated by ordinal logistic regression taking into account the following variables: age, sex, baseline National Institutes of Health Stroke Scale score (NIHSS), diabetes mellitus, arterial hypertension, history of prior ischemic stroke, known atrial fibrillation, anticoagulation treatment prior to stroke, intravenous thrombolysis and endovascular treatment. The abovementioned variables and interventions were included in the model. As this study focused on disability, only surviving patients were analyzed (mRS 0–5). The adjusted common odds ratio for NIHSS at hospital admission was analyzed as the secondary outcome. Common odds ratios are reported with 95%-confidence intervals to indicate statistical precision. Univariate analysis was performed using the Mann–Whitney U-test or χ^2^-test, as appropriate. *p*-values were corrected for multiplicity of testing using a false-discovery rate (FDR) with the Benjamini–Hochberg procedure. The significance level was set to *p* < 0.05, and all tests of hypotheses were two-sided. Data was analyzed with SPSS 26 (IBM; Armonk, NY, USA).

## 3. Results

### 3.1. Baseline Characteristics

In total, 141,287 patients hospitalized for acute ischemic stroke between 2010 and 2019 were analyzed (Table 1). The number of stroke patients documented per year (2010: 13,821; 2019: 14,210) and average age (2010: 73.6 ± 12.7 years; 2019 73.6 ± 12.9 years) showed only marginal variations during the study period. Thrombolysis rates increased from 10% in 2010 to 19% in 2019 (Appendix A). Thrombectomy rates increased from 0% in 2010 to 8% in 2019 (Appendix A). Treatment with anticoagulants prior to stroke increased from 9% in 2010 to 17% in 2019 (Appendix A). The mean duration of hospital stay did not change over time (2010: 9.2 ± 7.3 days; 2019: 9.1 ± 7.7 days). During the study period, the population in Hesse increased from 6.06 Mio (2010) to 6.29 Mio (2019).

### 3.2. Stroke-Associated Disability

The absolute number of severely disabled (mRS 4–5) stroke patients at hospital discharge decreased over time (2010: 3223 (equivalent to 53/100,000 population); 2019: 2429 (39/100,000 population). In individual years, however, there was a slight temporary increase, which was revised again in the long term. In parallel, the cumulative number of mRS points at discharge (as a measure of the burden of stroke disability to bear for the caregiving system) decreased over time (2010: 28,566 (471/100,000 population); 2019: 25,769 (410/100,000 population), OR 0.98 (95% CI 0.98–0.99), *p* < 0.001) (Appendix A). This trend was prominent for patients aged 80 years and older (OR 0.97 (95% CI 0.96–0.97), *p* < 0.001 Appendix A) and for patients with atrial fibrillation (OR 0.97 (95% CI 0.96–0.97), *p* < 0.001, Appendix A) but absent in younger stroke patients (OR 0.99 (95% CI 0.98–1.00), *p* < 0.001, Appendix A). The odds at hospital admission for a higher mRS at discharge decreased significantly by 3.7% per year (OR 0.96 (95% CI 0.96–0.97), *p* < 0.001). Additional inclusion of patients with mRS of 6 (i.e., mRS 4–6) did not change these effects. In addition, the items atrial fibrillation (OR 1.48 (95% CI 1.46–1.51), *p* < 0.001) and history of prior stroke (OR 1.51 [95% CI 1.48–1.54], *p* < 0.001) had detrimental effects on the functional outcome (mRS) at discharge. The recanalizing therapies thrombolysis (OR 0.72 (95% CI 0.70–0.74)) and endovascular treatment (OR 0.23 (95% CI 0.22–0.25)), if performed, strongly improved chances on a favorable mRS at discharge (Table 2).

The absolute number of severely affected stroke patients at hospital admission (NIHSS > 15) decreased over time (2010: 1589 (26/100,000); 2019: 1185 (19/100,000)). In parallel, the cumulative number of NIHSS points (as a measure of the burden of stroke severity to bear for the treating hospitals) decreased over time (2010: 89,285 [1472/100,000]; 2019: 77,673 [1235/100,000], OR 0.97 (95% CI 0.97–0.97), *p* < 0.001, Appendix A). This trend was prominent for patients aged 80 years and older (OR 0.96 (95% CI 0.95–0.96), Appendix A) and for patients with atrial fibrillation (OR 0.94 (95% CI 0.94–0.95), *p* < 0.001, Appendix A) but less strong in younger stroke patients (OR 0.98 (95% CI 0.98–0.99), *p* < 0.001, Appendix A). The odds for a higher NIHSS score at hospital admission decreased by 3.8% per year (OR 0.96 (95% CI 0.96–0.97), *p* < 0.001). The strongest predictor of an increased NIHSS at admission was the presence of atrial fibrillation (OR 1.66 (95% CI 1.63–1.70)) (Table 3).

## 4. Discussion

Our analysis demonstrates that stroke-associated disability in the population decreased over time from 2010 to 2019, both in terms of the absolute number of patients discharged with severe physical limitations (mRS 4–5) per year and in terms of the number of cumulative mRS points at hospital discharge per year (as a measure of disability burden for the caregiving system).

Whereas large-scaled epidemiological studies based on heterogeneous data sources have reported that stroke-associated mortality and disability adjusted life years (DALY) are declining in most countries [11], we demonstrated the reduction of stroke severity and stroke-associated disability over time in a defined population with a homogeneous health care system, taking into account both first-ever and recurrent strokes.

In 2019, the caregiving system in Hesse had to deal with 25% less severely disabled stroke patients (mRS 4–5) discharged from hospital than in 2010. Comparatively, in 2019, stroke centers in Hesse had to deal with 25% less severely affected stroke patients (NIHSS > 15) at hospital admission than in 2010. We found these effects strongly prominent in elderly patients and in patients with atrial fibrillation, but vastly absent in younger stroke patients. In view of the increased use of anticoagulants in the population, it is plausible that the prevention of severe cardioembolic strokes in elderly patients is largely responsible for the overall reduced stroke severity at hospital admission and—consequently—for the reduced number of severely disabled patients at discharge. Other preventive measures, such as statin therapy and anti-hypertensive medication, may also contribute to this effect. Only recently, an analysis focused on stroke incidence and mortality from a Danish registry from 2005 to 2018 showed a similar trend with a steady incidence of ischemic stroke in younger adults, while incidence dropped in adults above the age of 70, especially driven by a reduction of severe ischemic strokes [12]. Though the authors did not analyze NIHSS and mRS data, the findings are well comparable and largely strengthened by the results of our analyses.

It is noteworthy that this observational registry study did not comprise an exhaustive variety of potential confounding risk factors (e.g., lifestyle changes). Furthermore, the current study did not consider the occurrence of post-stroke depression or post-stroke pneumonia, both of which are known to have impactful detrimental effects on stroke recovery and disability [13,14,15]. The findings could, moreover, be biased by an artefactual reduction of stroke severity due to the more frequent use of enhanced diagnostic methods (e.g., MRI), allowing better identification of minor stroke. However, the total number of annual strokes was constant during the observational period, as was the annual number of young stroke patients <60 years. We, therefore, conclude that it is very plausible that the reported effect is likely associated with preventive measures, and not simply a consequence of an overall decreased stroke incidence or a raising of awareness towards milder strokes. Additionally, the successive implementation of recanalizing therapies may have contributed to disability reduction at discharge [10]. However, in our dataset, a maximum of 19% of patients per year received thrombolysis, and endovascular therapy was applied to a maximum of 8% of patients per year. This limits the overall effect of therapies on stroke-associated disability in the population. Furthermore, the missing disability reduction in younger stroke patients (in whom therapeutic measures are usually not withheld) speaks against a significant effect of improved treatment compared to the effect of reduced stroke severity at hospital admission. In addition, already prior to the wide establishment of thrombectomy data from the South London Stroke Register has shown a similar trend with a decrease in functional dependence, especially in patients above 55 years and with an underlying mechanism of cardio-embolism [16].

The logistic regression revealed that both atrial fibrillation as well as anticoagulation are predictive of an unfavorable outcome as measured by mRS, which at first glance contradicts the authors’ hypothesis of improved stroke prevention in the elderly. However, as atrial fibrillation as a risk factor is commonly associated with large-vessel occlusion and highly decreased functional outcome, and anticoagulation can only reduce said risk, it is plausible that both still depict a detrimental variable for the outcome parameter mRS.

Noteworthily, since 2013, an emergency alert and transportation system (IVENA, www.ivena-hessen.de, accessed on 12 September 2022) has been implemented in the federal state of Hesse with the aim of making the coordination of hospitals, rescue services, and the control centers in the care of emergency patients more effective and transparent. This measure might have contributed to faster hospital admissions of stroke patients, thus enabling earlier and more effective treatment options. Furthermore, in 2015, a network-wide, peer-to-peer acute stroke workflow improvement program was implemented within a considerable number of stroke units in the state of Hesse, aiming to improve the outcomes of acute recanalization therapies [17]. However, these measures affected all patients alike and especially an improvement during the hospital stay, while this study demonstrated that NIHSS at admission declined over time (e.g., prior to acute treatment) and that there was a specifically positive trend in elderly patients above 80 years of age.

The number and percentage of severely disabled stroke patients has declined in the state of Hesse from the year 2010 to 2019. The findings of this study implicate that recanalizing treatments can be impactful for the individual patient, but their effect in reducing the burden of stroke in the population is limited by the small number of eligible patients. We hypothesize that preventive measures currently have a stronger positive impact on the healthcare system, especially in elderly patients. Consequently, the results of this study should drive and guide future research on the widened implementation of primary and secondary stroke prevention in the population. Furthermore, further research on a more generous selection of patients eligible for acute recanalizing treatments is needed, e.g., provided by improved imaging protocols [18]. In addition, diagnostic biomarkers such as glial fibrillary acidic protein (GFAP) could be helpful in the selection of patients with a preferred outcome and to better predict their prognosis, though the evidence is outstanding [19,20]. Furthermore, the earlier detection of stroke patients, their faster referral to the hospital, and the earlier detection of patients with so-far silent brain infarcts [21,22] warrant further research. While prevention and acute stroke treatments aim to prevent brain ischemia, neuroprotective interventions for ischemic stroke have a large potential, though the lack of research is still significant [23,24]. Recent advances in stroke rehabilitation have also considerably reduced the stroke burden on the healthcare system, but more large stroke rehabilitation trials are needed to discriminate effective from ineffective rehabilitation strategies [25].

## 5. Conclusions

In summary, stroke-associated disability in the population has declined over the past decade. We hypothesize that preventive measures in the elderly may be a major driver for this effect.

## Figures and Tables

**Table 1 jcm-11-06942-t001:** Baseline characteristics and process measures.

	Overall
Baseline characteristics (*n*, %)	141,287
Age (years, mean, SD)	73.74 ± 12.87
≥80	37,752 (26.7)
<60	24,760 (17.5)
Female	67,507 (47.8)
NIHSS admission (median, IQR)	4 (2–8)
Need for care prior to stroke ^a^ (*n*, %)	
- Independent	46,217 (32.7)
- Care at home	19,248 (13.6)
- Institutional care	4939 (3.5)
Admission to hospital (*n*, %)	
- Self-initiated	15,141 (10.7)
- Via family physician	19,248 (13.6)
- Via emergency service	94,489 (66.9)
- Secondary transfer	12,409 (8.8)
OAT prior to admission (*n*, %)	
- No OAT	122,603 (86.8)
- Vitamin K antagonist	12,049 (8.5)
- DOAC	6635 (4.7)
Risk factors (*n*, %)	
- Previous stroke	33,086 (23.4)
- Hypertension	115,752 (81.9)
- Diabetes	38,242 (27.1)
Treatment (*n*, %)	22,480 (15.9)
- Thrombolysis only	16,603 (82.0)
- Endovascular treatment only	1538 (7.6)
- Combined	2107 (10.4)

^a^ data not available on all patients. Abbreviations: IQR (interquartile range), OAT (oral anticoagulation therapy), DOAC (dual oral anticoagulant).

**Table 2 jcm-11-06942-t002:** Secondary analysis: odds at hospital admission for a higher mRS at discharge.

Variable	Common Odds Ratio (95% CI)	*p*-Value
Impact of one year	0.96 (0.96–0.97)	<0.001
Age	1.04 (1.04–1.04)	<0.001
Female	1.22 (1.20–1.24)	<0.001
History of stroke	1.51 (1.48–1.54)	<0.001
Atrial fibrillation	1.48 (1.46–1.51)	<0.001
Prior anticoagulation	1.10 (1.06–1.14)	<0.001
Hypertension	1.02 (0.99–1.04)	<0.001
Diabetes	1.31 (1.28–1.34)	<0.001
Thrombolysis	0.72 (0.67–0.74)	<0.001
Endovascular treatment	0.23 (0.22–0.25)	<0.001

Abbreviations: NIHSS = National Institutes of Health Stroke Scale; CI = confidence interval.

**Table 3 jcm-11-06942-t003:** Secondary analysis: odds for a higher NIHSS at hospital admission.

Variable	Common Odds Ratio (95% CI)	*p*-Value
Impact of one year	0.96 (0.96–0.97)	<0.001
Age	1.02 (1.02–1.02)	<0.001
Female	1.22 (1.19–1.24)	<0.001
History of stroke	1.28 (1.25–1.31)	<0.001
Atrial fibrillation	1.66 (1.63–1.70)	<0.001
Prior anticoagulation	1.21 (1.17–1.26)	<0.001
Hypertension	0.98 (0.96–1.00)	n.s.
Diabetes	1.11 (1.08–1.13)	<0.001

Abbreviations: NIHSS = National Institutes of Health Stroke Scale; CI = confidence interval; n.s. = not significant.

## Data Availability

The data presented in this study are available upon reasonable request from all authors.

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
