# Peer review of "Reduction of Ischemic Stroke Associated Disability in the Population: A State-Wide Stroke Registry Analysis over a Decade"

_jcm, 2022, doi:10.3390/jcm11236942_

Round 1
Reviewer 1 Report
GENERAL COMMENTS
This monocentric, retrospective study provides important evidence proving that disability associated with ischemic stroke at patients’ discharge as well as admission has been decreasing in the last decade, in particular in elderly patients and in those with atrial fibrillation.
While there are some limitations to this work (e.g. the potential artefactual reduction in stroke severity due to identification of minor strokes thanks to better imaging), authors correctly address them in the discussion, and the findings are in line with epidemiological evidence.
Only minor points need to be addressed before the article is suitable for publication, concerning mainly the methodological distinction between correlation and causation, and discussion of literature, implications and perspective of the present research, to better valorize the very relevant evidence that authors describe.
ABSTRACT
1) The abstract is overall well-structured and compelling, thank you.
TITLE AND INTRODUCTION
1) The manuscript reads “Aim of this study was to assess whether the successive implementation of improvements in acute care and stroke prevention into routine clinical practice noticeably reduces stroke-associated disability in the population over time.” (line 45). However, the study is an observational study and was not designed to test the effect of “improvements in acute care and stroke prevention” on “stroke-associated disability” and causation cannot be inferred, although obviously hypotheses can be formulated in the Discussion (as the authors correctly did). A randomized controlled trial would be necessary to prove that the observed effects are due to certain interventions and that there is no spurious correlation. This paragraph in the Introduction should thus be reformulated specifying that a simple correlation analysis is performed, e.g. something like: “Aim of this study was to assess whether the stroke-associated disability in the population decreased over time.”.
2) For the same reason as of point 1, the title should be modified because it misleadingly suggests that the article proves that “Reduction of ischemic stroke associated disability in the population” is caused by “improved stroke prevention”. For instance, I suggest a title like “Reduction of ischemic stroke associated disability: a monocentric study over a decade”.
3) While the references are appropriate and well-chosen, the Introduction should detail additional previous research on this topic and better argument the relevance of the gap of knowledge.
METHODS
1) While data were acquired from a previous prospective study, this is a retrospective study, since data were acquired before the design of the present analysis, that was performed a posteriori. This should be clarified.
2) The “large prospective inpatient stroke registry” (Line 50) name or identification should be provided.
3) Correction for multiple comparisons should be performed and documented where appropriate.
RESULTS
1) Throughout the manuscript, a “decrease over time” of NIHSS and mRS is reported, however, each time (e.g. line 95) only the comparison between 2010 and 2019 is provided. Indeed, supplementary figures show that between 2010 and 2019 there were (non-significant?) increases in disability in some of the years compared to 2010, or decreases compared to 2019. While this is probably normal statistical noise, it should be mentioned and briefly addressed, since the manuscript now seems to suggest a uniform and steady decrease over time. Finally, the results of regression analysis between the raw disability scores (admission, discharge, and/or difference between the two) and time should be provided (also as a supplementary result) to further support the findings.
2) Any other non-significant analysis that the author performed should be mentioned in the supplementary results to offer a more balanced viewpoint on the evidence.
3) Similarly to point 1, lines 96-100 read: “In parallel, the cumulative number of mRS points at discharge (as a measure of the burden of stroke disability to bear for the caregiving system) decreased over time (2010: 28,566 [471/100,000 population]; 2019: 25,769 [410/100,000 population]) (Suppl. fig. 3). This trend was prominent for patients aged 80 years and older (Suppl. fig. 4) and for patients with atrial fibrillation (Suppl. fig. 5) but absent in younger stroke patients (Suppl. fig. 6).”.
Suppl. fig. 5 does not visually show a clear trend, and cumulative mRS scores in 2013 and 2014 were lower than 2019, while 2017 and 2018 had higher mRS scores than 2010. How do the authors explain these elements that seem to contradict their hypotheses? Are these differences significant?
4) The p-values should be provided in the Tables where appropriate.
DISCUSSION
1) As mentioned above, a better discussion of existing literature and in particular of perspectives and implication of this findings is needed. These are important findings, what makes them relevant? How could they drive or guide further research (e.g. into novel therapeutic strategies or better early stroke diagnosis)? For instance, at lines 155-156, the statement “This limits the overall effect of therapies on stroke-associated disability in the population” is very relevant and interesting, authors should further discuss this point, mentioning the lack of research 1) into potential neuroprotective interventions for ischemic stroke, and 2) into potential diagnostic or physiopathological biomarkers, to improve brain ischemia prognosis. Recent studies on these topics should be mentioned as potential perspectives, including (but not necessarily limited to):
- Neuroprotection in Acute Ischemic Stroke: A Battle Against the Biology of Nature (DOI: https://doi.org/10.3389/fneur.2022.870141)
- Emerging neuroprotective strategies for the treatment of ischemic stroke: An overview of clinical and preclinical studies (DOI: 10.1016/j.expneurol.2020.113518)
- Remote Ischemic Conditioning in Ischemic Stroke and Myocardial Infarction: Similarities and Differences (DOI: 10.3389/fneur.2021.716316)
- Blood Biomarkers for Stroke Diagnosis and Management (DOI: 10.1007/s12017-019-08530-0)
- Platelet, Plasma, Urinary Tryptophan-Serotonin-Kynurenine Axis Markers in Hyperacute Brain Ischemia Patients: A Prospective Study (DOI: https://doi.org/10.3389/fneur.2021.782317)
- Blood Biomarkers for Stroke Diagnosis and Management (DOI: 10.1007/s12017-019-08530-0)
- Acute ischemic stroke biomarkers: a new era with diagnostic promise? (DOI: 10.1002/ams2.696
CONCLUSION
1) For the same reasons detailed in comment 1 to the Introduction, the sentence “In summary, stroke-associated disability in the population is declining, and preventive measures in the elderly are a plausible driver for this effect.” (lines 170-171) should be toned down, for instance: “In summary, stroke-associated disability in the population is declining. WE HYPOTHESIZE that preventive measures in the elderly MAY BE a plausible driver for this effect.”
MINOR COMMENTS
1) I suggest to avoid writing “Same as figure…” (e.g. in the captions to Supplemental Figures 8-10) and to repeat and/or adapt the figure caption each time for better readability.
Reviewer 2 Report
Current reports emphasize that the proportion of post-stroke patients with disability is decreasing in the general population. Risk factors associated with disability after stroke include age, neurological deficits, cognitive function, depression and social support. Also, analysis of effective prevention strategies and acute stroke therapy reduce the global burden of stroke.
Reviewer 3 Report
This work explores the possible relations between stroke associated disability in a defined population and stroke prevention strategies. Authors demonstrated the reduction of stroke severity and stroke-associated disability among 141,287 patients with ischemic stroke, admitted between 2010 and 2019. Despite the limitations listed (lack of information about lifestyle changes, risk factors), results showed a relevant effect of prevention therapies, in addition to the implementation of recanalizing treatment.
